# Substantially enhanced homogeneous plastic flow in hierarchically nanodomained amorphous alloys

Ge Wu [1] ✉, Sida Liu[2], Qing Wang[3], Jing Rao[4], Wenzhen Xia[5], Yong-Qiang Yan[1], Jürgen Eckert[6,7], Chang Liu [8] ✉, En Ma [8] & Zhi-Wei Shan[1]

To alleviate the mechanical instability of major shear bands in metallic glasses at room temperature, topologically heterogeneous structures were introduced to encourage the multiplication of mild shear bands. Different from the former attention on topological structures, here we present a compositional design approach to build nanoscale chemical heterogeneity to enhance homogeneous plastic flow upon both compression and tension. The idea is realized in a Ti-Zr-Nb-Si-XX/Mg-Zn-Ca-YY hierarchically nanodomained amorphous alloy, where XX and YY denote other elements. The alloy shows ~2% elastic strain and undergoes highly homogeneous plastic flow of ~40% strain (with strain hardening) in compression, surpassing those of mono- and heterostructured metallic glasses. Furthermore, dynamic atomic intermixing occurs between the nanodomains during plastic flow, preventing possible interface failure. Our design of chemically distinct nanodomains and the dynamic atomic intermixing at the interface opens up an avenue for the development of amorphous materials with ultrahigh strength and large plasticity.

Homogeneous plastic flow, different from shear banding, is a ductile deformation mode of metallic glasses (MGs) when the sample size is reduced to sub-100 nm[1,2]. Computational simulations have shown their advantages in predicting nanostructures of MGs with homogeneous plastic flow. Evolutionary algorithms[3] are able to search for MG candidates with excellent glass forming ability (GFA), and thus optimize the existing MGs. Machine learning based modelings[4–6] can predict structure heterogeneities for different MG systems and correlate their structures with mechanical properties. Multiscale modelings by molecular dynamics simulations[7,8] demonstrate that heterogeneous MGs (nanoglasses or nanolaminated composites)

have enhanced homogeneous deformation. In experiments, however, it is difficult to achieve very large heterogeneity as that in simulations and the mechanical strain rates are much lower (~$10^{-3}$ s$^{-1}$ vs. ~$10^7$ s$^{-1}$). Adding crystalline phase into the amorphous matrix[9,10] is an efficient experimental approach to enhance plasticity, resulting from dislocation movement inside the crystalline phase and multiple shear banding events in the amorphous matrix. By reducing the size of the amorphous phase to sub-10 nm in the crystal-glass alloys[11–13], confined plastic flow of the nano-sized amorphous phase can be realized, due to intrinsic small size and much different mechanical responses of the crystalline and amorphous phases. Therefore, here we conceive

[1]Center for Advancing Materials Performance from the Nanoscale (CAMP-Nano) and Hysitron Applied Research Center in China (HARCC), State Key Laboratory for Mechanical Behavior of Materials, Xi'an Jiaotong University, 710049 Xi'an, China. [2]Institute for Advanced Technology, Shandong University, 250061 Jinan, China. [3]Laboratory for Microstructures, Institute of Materials, Shanghai University, 200072 Shanghai, China. [4]Max-Planck-Institut für Eisenforschung, Max-Planck-Straße 1, Düsseldorf 40237, Germany. [5]School of Metallurgical Engineering, Anhui University of Technology, 243000 Maanshan, China. [6]Erich Schmid Institute of Materials Science, Austrian Academy of Sciences, Jahnstraße 12, Leoben A-8700, Austria. [7]Department of Materials Science, Chair of Materials Physics, Montanuniversität Leoben, Jahnstraße 12, Leoben A-8700, Austria. [8]Center for Alloy Innovation and Design (CAID), State Key Laboratory for Mechanical Behavior of Materials, Xi'an Jiaotong University, 710049 Xi'an, China. ✉e-mail: gewuxjtu@xjtu.edu.cn; chang.liu@xjtu.edu.cn

the idea of introducing nanodomains with large mechanical differences in fully amorphous alloys, in order to realize confined plastic flow with no mature shear bands in the nanodomains. Compositional design for each amorphous nanodomain can introduce large mechanical differences[14]. However, in order to achieve separated amorphous domains in the as-fabricated material (Fig. 1a), positive mixing enthalpy is required for certain constituent elements[15], which actually degrades GFA. This indicates that the conventional top-down fabrication method, usually including quenching and post-annealing, may favor crystallization. We thus use a bottom-up fabrication method to alternately deposit pre-designed nanodomains. Magnetron sputtering depositions with template substrates[16] or co-sputtering depositions via high throughput method[17] are efficient to fabricate materials with a compositional gradient in a horizontal direction, and thus different microstructures can be achieved. These strategies show their advantages in the development of compositionally complex crystalline[18] and amorphous[19] alloys. The purpose of the current study is to fabricate materials with hierarchically nanodomained compositions in both horizontal and vertical directions, which is different from the above strategies. The constituent elements in each nanodomain have large negative mixing enthalpy to guarantee an amorphous structure, while the principal constituent elements from the two nanodomains have positive mixing enthalpy, reducing the driving force for interdiffusion[20]. Although positive mixing enthalpy is required for the principal constituent elements of the neighboring amorphous nanodomains, the value of the mixing enthalpy should not be large, which sets an alloy design guideline. The hierarchically nanodomained amorphous alloy reveals atomic intermixing during plastic deformation, as shown in the context below. In this circumstance, the deformed alloy will suffer from potential crystallization if the positive mixing enthalpy of the principal constituent elements were large. Therefore, small positive mixing enthalpy is needed. Here, the value of small positive mixing enthalpy should be in the range of 5–20 kJ/mol, as suggested in ref. [15]. We summarize mixing enthalpy of atomic pairs[21] from frequently used metallic elements (Supplementary Table 1), and highlight the small positive values in green color, suggesting potential principal constituent elements for the neighboring amorphous nanodomains. For example, Ti and Mg have a small positive mixing enthalpy of 16 kJ/mol, and thus Ti-based and Mg-based amorphous alloys are good candidates for the neighboring amorphous nanodomains.

Here, we realize this idea by using magnetron sputtering to alternately deposit 3.4-nm-thick Ti-Zr-Nb-Si-*XX* and 2.9-nm-thick Mg-Zn-Ca-*YY* nanodomains using $Ti_{60}Zr_{10}Nb_{15}Si_{15}$ (at.%) and $Mg_{60}Zn_{35}Ca_5$ (at.%) targets, respectively. The received material shows a high yield strength of 1.6 GPa and plastic deformation of over 60% strain (including ~40% homogeneous plastic flow). During plastic flow, dynamic atomic intermixing occurs between the amorphous nanodomains, originating from atomic rearrangements via profuse shear transformation events. This behavior prevents possible interface failure and thus leads to mechanical stabilization of the whole material.

## Results

### Enthalpy-guided alloy design

It is worthwhile noting that severe interdiffusion (>50 at.%) occurs in sputtered nanolaminates when the nanolayer thickness is reduced to ~3 nm[22]. The thermodynamic driving force is large negative mixing enthalpy of the constituent elements[20] in the adjoining nanolayers. To reduce this effect, here we used Mg and Ti with positive mixing enthalpy, as the principal constituent elements of the nanodomains. It is reported that hierarchical herringbone-like microstructure is efficient to provide crack buffering effect in a crystalline high-entropy alloy[23]. Here we introduced structure/composition hierarchy into amorphous materials, exploiting the different growth modes of the nanodomains. We noted that the as-deposited Mg-Zn-Ca alloy has a nano-dual-phase amorphous structure, where Ca-enriched amorphous filaments with a diameter of ~5 nm are aligned in a Mg-enriched amorphous matrix from the growth direction (Supplementary Fig. 1). We thus used Mg-Zn-Ca amorphous nanolayer as a precursor to realize skeleton-like hierarchical nanostructure (Fig. 1b) of the whole material, distinct from multilayer structure of conventional films fabricated also by alternate deposition[24]. Both the two domains are amorphous, as indicated by the halo ring feature (inset, Fig. 1c) in selected-area electron diffraction (SAED). Figure 1d, e shows that the Ti-Zr-Nb-Si-*XX* and Mg-Zn-Ca-*YY* domains are bright and dark regions in aberration-corrected high-angle-annular dark-field (HAADF) scanning transmission electron microscopy (STEM) image (Fig. 1c), respectively. The compositions of the alternately sputter-deposited amorphous domains are $Ti_{55}Zr_5Nb_6Si_{15}Mg_6Zn_{10}Ca_3$ (at.%) and $Mg_{37}Zn_{35}Ca_3Ti_{15}Zr_2Nb_2Si_6$ (at.%). The side-view TEM image of the material shows a wavy structure, which is vertically separated by Mg, Zn and Ca-enriched regions with a thickness of ~5 nm and a wavelength of ~65 nm.

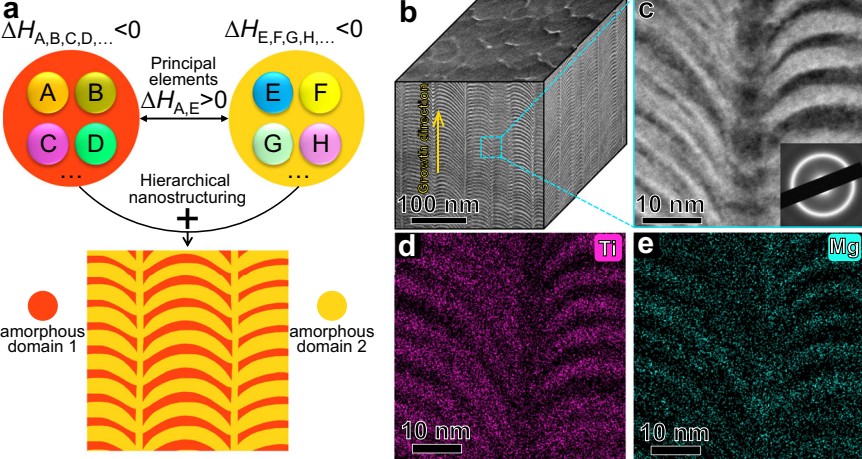

**Fig. 1 | Enthalpy-guided alloy design strategy for the hierarchically nanodomained amorphous alloy. a** Schematic presentation of the alloy design strategy. A, B, C, D, E... are constituent elements of the amorphous domains, where the principal elements (A and E) of the two domains have positive mixing enthalpy ($\Delta H_{A,E} > 0$), and the constituent elements of each domain have negative mixing enthalpy ($\Delta H_{A,B,C,D,...} < 0$, $\Delta H_{E,F,G,H,...} < 0$). **b** Typical bright-field plan-view and side-view TEM images. **c** Enlarged HAADF-STEM image, probed from the dashed square area in (**a**). The inset shows a corresponding SAED pattern. **d, e** EDS mapping of Ti and Mg from the side-view probing.

Correspondingly, the plan-view TEM image (Fig. 1b) shows ~65-nm-diameter amorphous granular regions surrounded by ~5-nm-thick amorphous intergranular layers. The wavy structure shown in the side-view images (Fig. 1b, c) results from the columnar growth of nanolaminates[25]. As a comparison, the sputtered Ti-Zr-Nb-Si reference amorphous alloy shows a homogeneous composition without the wavy structure (Supplementary Fig. 2).

## Mechanical properties

We carried out micro-compression tests on micro-pillars fabricated from the materials to investigate their mechanical responses at room temperature. The compression experiments were conducted on Mg-Zn-Ca amorphous alloy, Ti-Zr-Nb-Si amorphous alloy and the hierarchically nanodomained amorphous alloy pillar samples with a diameter of 1 μm and height of 2 μm to avoid sample size effect[2,26]. The Mg-Zn-Ca and Ti-Zr-Nb-Si reference amorphous alloys have yield strength of 1.0 GPa and 2.2 GPa, respectively, and identical elastic strain of ~2% (Fig. 2a). Their stress-strain curves reveal large serrations after yielding, which are related to the generation and propagation of shear bands. The mechanical responses and deformed pillar morphologies (Fig. 2b, c) of the reference amorphous alloys show a brittle deformation mode, consistent with other micro-compression reports for MGs[27–29]. In contrast, the hierarchically nanodomained amorphous alloy reveals a yield strength of 1.6 GPa and plastic deformation of over 60% strain (including ~40% homogeneous deformation) (Fig. 2a). The pillar compression tests under identical conditions were repeated for at least three times to ensure statistical significance (Supplementary Fig. 3). Figure 2e shows improved mechanical property of the hierarchically nanodomained amorphous alloy compared with that of other amorphous alloys with similar sample dimensions[27–30]. The yield strength is normalized by Young's modulus ($\sigma_y/E$). The normalized yield strength of all samples lies in the regime typical for MGs ($\sigma_y = E/50$)[14]. However, the plastic flow of the hierarchically nanodomained amorphous alloy is substantially more homogeneous, which is usually seen in the supercooled liquid but with low normalized yield strength. The deformed pillar after >60% strain shows a relatively homogeneous deformation feature with only a few small shear bands (Fig. 2d). We checked the morphology of the deformed pillar with ~30% strain, and it shows no shear bands (inset, Fig. 2a). Therefore, the stress deviation (apart from the green dashed line on Fig. 2a) and small serrations on the stress-strain curve after 40% strain correspond to multiple shear banding events.

The apparent stress increase after pop-in events for the reference amorphous alloys results from a sudden increase of pillar diameter by shear banding but does not represent strain hardening. However, the stress-strain curve of the hierarchically nanodomained amorphous alloy reveals a continuous deviation from the linear elastic region as the stress increases from 1.6 to 2.0 GPa (Fig. 2a), suggesting a strain hardening response. We further did compression experiments on nanopillars extracted from pre-deformed and un-deformed samples (Supplementary Fig. 4). The yield strength of the pre-deformed material is 1.8 GPa, 0.2 GPa higher than that of the un-deformed material, due to a change in chemical composition and structure (atomic intermixing between the two nanodomains).

It is reported that MGs can be rejuvenated to an elevated energy state by confined deformation[31], which facilitates the homogeneous plastic flow of ~1% strain (with strain hardening) in compression. It is known that the elevated energy[31] of MGs corresponds to ultra-fast cooling at ~$10^6$ K/s, three to four orders of magnitude higher than the possible cooling rate for a conventional cast rod, but that effective cooling rate is lower than what we used in magnetron sputtering deposition (~$10^{10}$ K/s)[32]. Therefore, it is supposed that the deposited hierarchically nanodomained amorphous alloy has an elevated energy. However, the as-deposited Mg-Zn-Ca and Ti-Zr-Nb-Si reference amorphous alloys show shear banding-dominated plastic deformation,

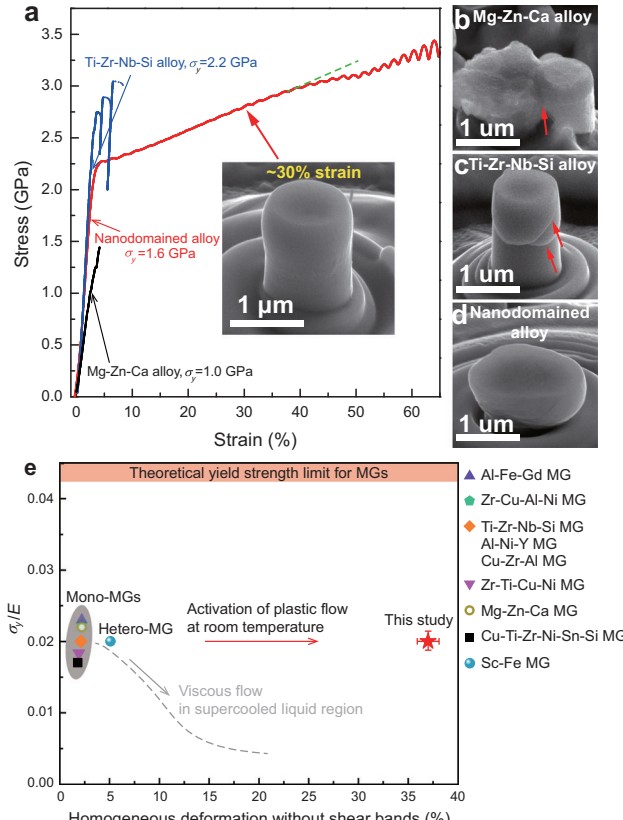

**Fig. 2 | Improved mechanical property of the hierarchically nanodomained amorphous alloy compared with that of reference mono- and hetero-MGs.** Here, mono- and hetero-MGs represent MGs with homogeneous and heterogeneous structures, respectively. **a** Compressive engineering stress-strain curves of the pillar samples tested with identical conditions at room temperature. The height/diameter ratio is 2, and the diameter is 1 μm. The arrows mark the yield points of the samples. The inset is a SEM image of a compressed pillar (hierarchically nanodomained amorphous alloy) with ~30% strain, showing homogeneous plastic deformation without shear bands. **b**–**d** SEM images of the corresponding Mg-Zn-Ca amorphous alloy, Ti-Zr-Nb-Si amorphous alloy and hierarchically nanodomained amorphous alloy pillars after compression. Some of the shear bands in (**b**) and (**c**) are indicated by red arrows. **e** Yield strength normalized by Young's modulus ($\sigma_y/E$) vs. homogeneous deformation without shear bands of the hierarchically nanodomained amorphous alloy, tested in micro-pillar compression at room temperature, in comparison with other mono- or hetero-MGs. The Young's modulus of the hierarchically nanodomained amorphous alloy is 80 GPa, obtained from nanoindentation. Error bars denote the standard deviations of the $\sigma_y/E$ and strain values.

which therefore indicates that the large homogeneous plastic flow and the observed strain hardening of the hierarchically nanodomained amorphous alloy are not due to energy elevation.

## Deformation mechanism

We performed micro-indentation and pillar compression on the hierarchically nanodomained amorphous alloy and then conducted aberration-corrected STEM on the deformed materials to investigate the deformation mechanism. The period thickness of the Ti-Zr-Nb-Si-$XX$/Mg-Zn-Ca-$YY$ domains decreases from 6.3 to 5.5 nm (13% strain), as the strain increases. The deformed region with >70% strain shows low contrast between the amorphous domains in HAADF-STEM imaging (Fig. 3c), indicating the reduced compositional difference. The maze-like pattern in the BF-STEM image confirms the amorphous structure (Fig. 3d). Near-atomic resolution EDS mappings were performed on these regions with different strains. One-dimensional (1D) compositional profiles probed from the regions with strains from 0% to >70%

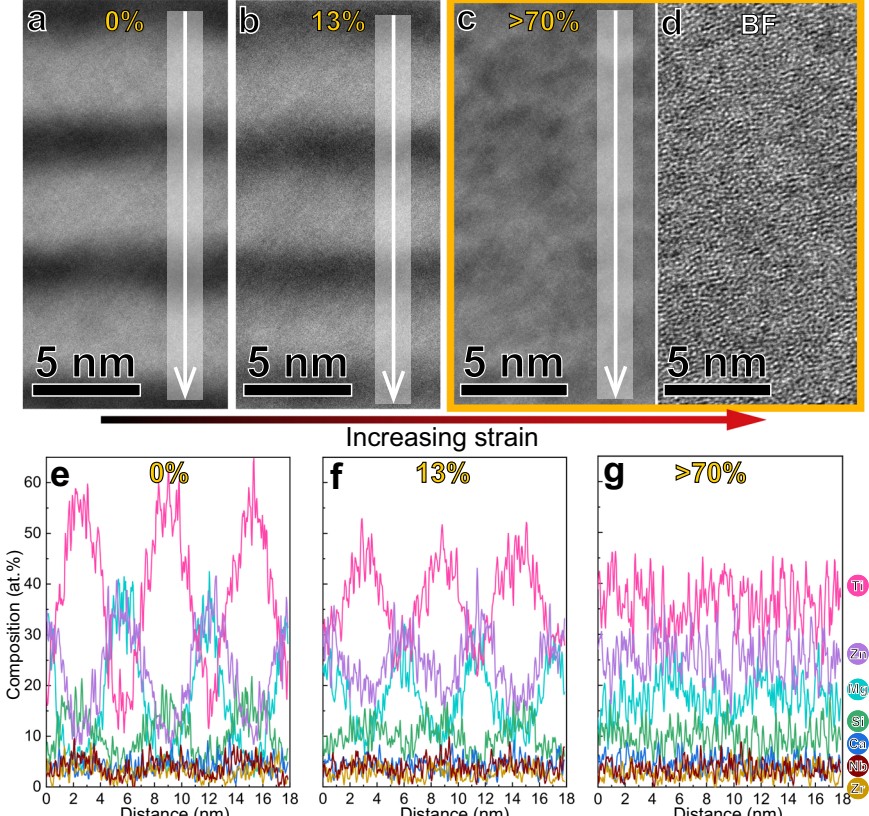

**Fig. 3 | Dynamic atomic intermixing of the hierarchically nanodomained amorphous alloy during plastic deformation. a–c** HAADF-STEM images of materials with different strains, probed from the as-deposited sample (0% strain), deformed region beneath an indent (Supplementary Fig. 5, 13% strain) and a compressed nanopillar (red curve sample, Supplementary Fig. 4, >70% strain). The strain in (**b**) was calculated based on the thickness reduction percentage of the amorphous domains. **d** BF-STEM image of the deformed region with >70% strain. The maze-like pattern confirms its amorphous structure. **e–g** 1D compositional profiles of the deformed regions indicated by the white arrows in (**a–c**), showing the amplitude of the elemental concentration difference across the nanodomains decreases with increasing strain.

show that the amplitude of the elemental concentration difference decreases as the strain increases (Fig. 3e−g). This indicates that dynamic atomic intermixing between the adjacent amorphous domains occurs during homogeneous plastic flow. It has been reported that the deformed MGs usually show elemental re-distribution inside shear bands, where specific element enriched and depleted regions both exist[33,34]. However, the hierarchically nanodomained amorphous alloy here shows a composition homogenization of the amorphous domains. The atomic intermixing is usually observed in highly deformed crystalline alloys[35,36], known as an externally driven mechanical alloying effect. It was found that dislocations drag highly concentrated atoms from one phase to another[35,36], inducing composition homogenization of the alloy during plastic deformation. Different from that in crystalline alloys, plasticity carriers in MGs are shear transformation (ST) zones[37,38] instead of dislocations. The homogeneous plastic flow of MGs is based on the activation of profuse ST events[39], which induce appreciable rearrangement of atoms between the adjacent amorphous nanodomains. This kinetic process facilitates homogenization of the nanodomains, rather than phase separation toward equilibrium reported inside some shear bands[34].

We use a schematic diagram to illustrate the advantages of the chemically distinct nanodomains and deformation-induced dynamic atomic intermixing mechanism (Fig. 4). The Young's modulus of the Mg-Zn-Ca-*YY* domain is expected to be smaller than that of the Ti-Zr-Nb-Si-*XX* domain, considering that the Young's modulus of the Mg-Zn-Ca and Ti-Zr-Nb-Si reference amorphous alloys is 45 GPa and 110 GPa, respectively, obtained from nanoindentation. Therefore, the profuse ST events occur earlier in the Mg-Zn-Ca-*YY* domain, and as the flow

stress increases, these events occur in the Ti-Zr-Nb-Si-*XX* domain afterward. The non-homogeneous deformation[40] of the two amorphous domains induces tiny serrated stress flow (from ~5 to ~40% strain, Fig. 2a) during homogeneous plastic flow. In this process, the dynamic atomic intermixing reduces compositional difference (Fig. 3) and in turn reduces Young's modulus mismatch between the two amorphous domains[14]. This decreases stress concentrations between the amorphous domains, preventing any potential interface failure. Furthermore, extrusion of softer phases during pillar compression, often observed in crystalline (softer)-amorphous (harder) nanolaminates[41], does not occur in the current material. This facilitates co-deformation of the amorphous domains with reduced plastic strain difference after the atomic intermixing. The dynamic atomic intermixing thus enhances the mechanical stability of the hierarchically nanodomained amorphous alloy during homogeneous plastic flow. The atomic intermixing between the amorphous domains becomes severe after the material undergoes large strains (Fig. 3c). Eventually, the composition or Young's modulus of the two amorphous domains would be homogenized to become similar (Fig. 3c, g).

## Tensile behavior

We further conducted in-situ tension experiments in an aberration-corrected TEM to reveal the tensile behavior of the hierarchically nanodomained amorphous alloy at room temperature (Fig. 5). The thickness of TEM lamella is ~50 nm before tension. The TEM lamella contains more than seven of the Ti-Zr-Nb-Si-*XX*/Mg-Zn-Ca-*YY* structural units in the thickness direction of the specimen and should reflect an average mechanical response of the whole material in tension. The

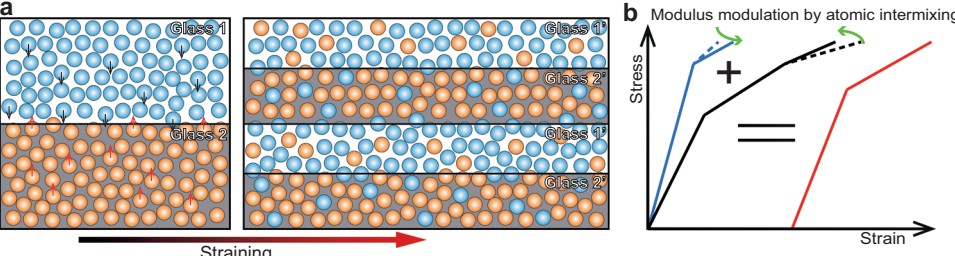

**Fig. 4 | Illustration of the dynamic atomic intermixing mechanism. a** The two amorphous domains "Glass 1" and "Glass 2" have different compositions and Young's modulus. During homogeneous plastic flow, dynamic atomic intermixing reduces compositional differences between the amorphous domains, impeding the formation of near-equilibrium phases. **b** Schematic stress-strain curves of the hierarchically nanodomained amorphous alloy (red curve) and its component amorphous domains (blue and black curves stand for Ti-Zr-Nb-Si-*XX* and Mg-Zn-Ca-*YY* domains, respectively). The atomic intermixing during plastic flow reduces the modulus difference between the two amorphous domains, decreasing the stress concentrations and preventing interface failures.

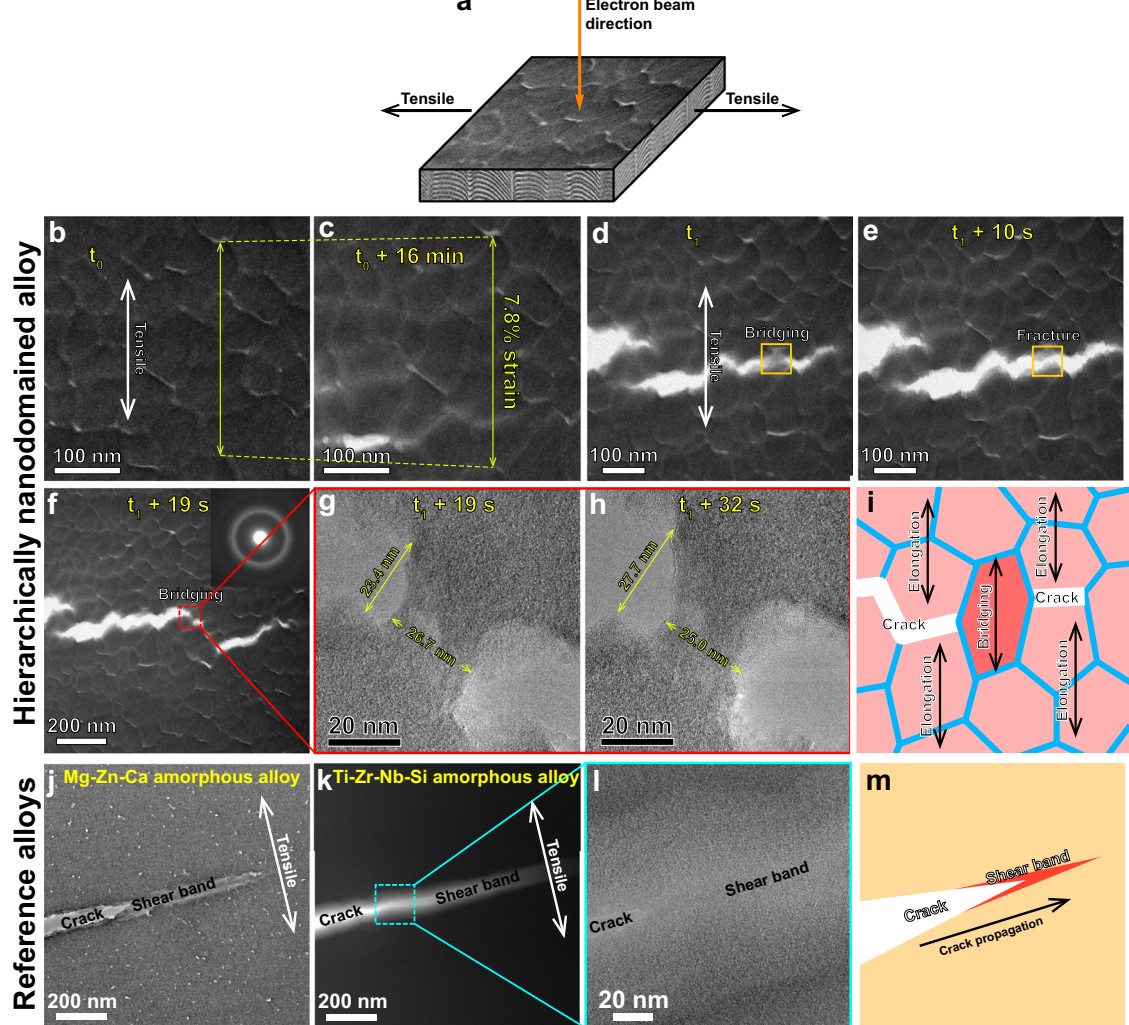

**Fig. 5 | In-situ TEM tensile deformation behavior of the hierarchically nano-domained amorphous alloy. a** In-situ TEM sample setup, showing electron beam (view) and tensile directions. **b**–**f** Bright-field TEM images of the hierarchically nanodomained amorphous alloy upon tension, **b**, **c** before and **d**–**f** during cracking. The instantaneous times are indicated by the yellow captions, where $t_0$ and $t_1$ are two initial times. The material undergoes 7.8% strain before cracking and reveals nano-bridging behavior for cracks during cracking. **g**, **h** High-resolution (HR) TEM images of the magnified area indicated in (**f**). The length and width of this area change from 23.4 nm to 27.7 nm and 26.7 nm to 25.0 nm, respectively, showing plastic flow behavior during tension. **i** Schematic illustration of the hierarchically nanodomained amorphous alloy upon tension. The materials undergo elongation before cracking. The cracks are generated from the amorphous intergranular shells, and their propagation can be impeded by the amorphous granular regions. These regions undergo homogeneous plastic flow subsequently, bridging the cracks. **j**–**l** Bright-field TEM images of the Mg-Zn-Ca and Ti-Zr-Nb-Si reference amorphous alloys during tension, showing cracking from the shear bands ahead. **m** Schematic illustration of the reference amorphous alloys upon tension, showing the propagation of a crack from the shear band.

material undergoes a large strain of 7.8% before the onset of cracking (Fig. 5c), indicating a stable flow behavior. A similar ductilization phenomenon by dual-amorphous phase has been achieved in a Co-Fe-Ta-B-O oxide glass[42], which shows an enhanced tensile plasticity of 2.7%. Furthermore, during cracking, the current hierarchically nano-domained amorphous alloy shows nano-bridging behavior for cracks and subsequent plastic flow of the bridging regions (Fig. 5d–h), in contrast to shear band induced cracking behavior of conventional MGs (Fig. 5j–l). Previous successful ductilization approaches for MGs have demonstrated that heterogeneous amorphous structures[43–45], embedded crystalline phases[9,10,46] and glass-glass interfaces[30] are able to deflect the propagation of shear bands, preventing brittle fracture dominated by the major shear band. In the current hierarchically nanodomained amorphous alloy, the amorphous granular regions not only deflect the crack propagation but also bridge the cracks by homogeneous plastic flow during tension. The homogeneous plastic flow behavior of the amorphous granular regions is expected from their small size, as MGs can undergo plastic flow without shear banding when the sample size is smaller than 100 nm[1,2]. Also, the design of the hierarchical nanostructure successfully realizes a buffering effect by nano-bridging of the tensile cracks, thanks to the extensive homogeneous flow.

In summary, we developed hierarchically nanodomained amorphous alloys with large chemical differences across Ti-Zr-Nb-Si-*XX* and Mg-Zn-Ca-*YY* amorphous nanodomains. Taking advantage of the intrinsic size (referring to the size of the structural unit) effect of amorphous domains at the nanoscale, the generation of shear bands is suppressed, which facilitates the activation of homogeneous plastic flow upon yielding. The material exhibits a yield strength of 1.6 GPa and ~40% homogeneous deformation, which surpasses most MGs. The high mechanical stability during homogeneous plastic flow results from dynamic atomic intermixing, which lowers the Young's modulus mismatch between the amorphous domains, thus preventing stress concentration-induced interface failure. These findings illustrate a nanostructuring approach for MGs, based on chemically distinct nanodomains and dynamic atomic intermixing during deformation. It is shown that this nanostructured design approach enables amorphous alloys with both ultrahigh strength and large deformability, which overcomes the brittleness of amorphous materials at room temperature. The strong and ductile nanodomained amorphous films may find broad applications, such as in load-bearing micro-electromechanical systems and flexible devices.

## Methods

### Fabrication of the materials

We used magnetron sputtering as the fabrication method. The background vacuum was $4 \times 10^{-7}$ Torr. $Mg_{60}Zn_{35}Ca_5$ (at.%) and $Ti_{60}Zr_{10}Nb_{15}Si_{15}$ (at.%) alloy targets (99.9 at.% purity) were used for alternate sputtering. The powers of the two targets were switched between on and off for 1 min during the alternate sputtering. The Ar working pressure was 3 mTorr, the substrate bias power was 180 W and the temperature of the substrate was below 50 °C. The samples with a thickness of ~2 μm were deposited on Si (1 0 0) substrates. The thickness of the films is large enough to fabricate micro-pillar samples by using a focused ion beam (FIB) facility. The reference Ti-Zr-Nb-Si and Mg-Zn-Ca amorphous alloys were prepared using magnetron sputtering with $Mg_{60}Zn_{35}Ca_5$ (at.%) and $Ti_{60}Zr_{10}Nb_{15}Si_{15}$ (at.%) alloy targets (99.9 at.% purity), respectively.

### Structural and compositional characterization

The microstructures of the alloys were investigated by using (S)TEM. We used a JEM 2100 F FEG TEM (from JEOL), operated at 200 kV, to do bright-field imaging and SAED. The plan-view and side-view TEM lamellae were prepared with a dual-beam FIB instrument (FEI Helios Nanolab 600). The final milling voltage/current was 2 kV/23 pA, which was sufficiently small to reduce the FIB damage. HR-STEM imaging and EDS were carried out using a 300 kV probe aberration-corrected FEI Titan Themis. For HAADF imaging, a probe convergence semi-angle of 23.8 mrad and inner and outer semi-collection angles ranging from 73 to 200 mrad were used. For BF-STEM imaging, inner and outer semi-collection angles from 13 to 21 mrad were used. For EDS mapping, the dwell time was 10 μs per pixel with a map size of 1024 × 1024 pixels; a complete process took 100 frames (21 min) of EDS mapping to reach an appropriately high signal-to-noise ratio. Needle-shaped specimens required for APT were fabricated by lift-outs and annular milled by FIB. The APT measurements were performed in a local electrode atom probe (CAMEACA LEAP 5000XR). The specimens were analyzed at 60 K in laser mode with a laser power of 20 pJ, a pulse repetition rate of 200 kHz, and an evaporation detection rate of 0.2% atom per pulse. Imago Visualization and Analysis Software (IVAS) version 3.8.8 was used for creating the 3D reconstructions and data analysis.

### Mechanical characterization

Micro-compression and tension were performed using a Hysitron TI950 nanoindenter and a PI95 PicoIndenter with a diamond punch and a W gripper, respectively under displacement-control mode and at a strain rate of ~$2 \times 10^{-3} s^{-1}$. The Hysitron TI950 nanoindenter is an ex-situ instrument in air, and thus the electron beam-induced composition and structure changes can be ruled out during compression. Micro-pillar and dog-bone-shaped samples were fabricated using FIB, with 30 kV/7 pA as the final milling condition. The aspect ratio (height/diameter) of the pillar was 2, and the taper angle of each pillar was less than 1.5°. The engineering stress $\sigma$ was calculated using $F/A_O$, where $F$ is the measured force and $A_O$ is the original cross-sectional area at the top of pillar samples or the cross-sectional area of dog-bone-shaped samples. The engineering strain $\varepsilon$ was calculated using $L/L_O$, where $L$ is the measured displacement and $L_O$ is the original length of the samples.

### In-situ TEM tension

The in-situ TEM tension experiments at room temperature were conducted by using a Gatan model 654 single-tilt straining holder with the normal e-beam density in an image aberration-corrected FEI Titan (Themis 80-300).

## Data availability

All data generated or analyzed during this study are included in the article and its Supplementary Information files, and are available from the corresponding authors upon request.

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

## Acknowledgements

G.W. acknowledges support from the National Natural Science Foundation of China (52271114). G.W. and C.L. acknowledge supports from National Natural Science Fund for Excellent Young Scientists Fund Program (Overseas). We thank D. Raabe, D. Ponge and J. Best at Max-Planck-Institut für Eisenforschung and J. Lu at the City University of Hong Kong for helpful discussions. We thank P.-C. Zhang, P. Zhang, Q.-Q. Fu and D.-L. Zhang at Xi'an Jiaotong University for technical support.

## Author contributions

C.L. designed the alloys and guided the project with Z.W.S.; C.L. and G.W. conducted FIB and TEM experiments; G.W. and C.L. conducted APT characterization and data analysis; C.L., G.W., S.L., J.R., W.X. and Y.Q.Y. conducted micro-compressions and micro-indentations; G.W. and C.L. conducted (S)TEM characterization and in-situ tension experiments; C.L., Q.W., J.E., E.M. and Z.W.S. contributed to the interpretations of the observations; G.W., C.L. and E.M. wrote the paper; all authors contributed to the discussion of the results.

## Competing interests

The authors declare no competing interests.
