## [Peer Review File · Nature Communications]

Substantially enhanced homogeneous plastic flow in hierarchically nanodomained amorphous alloysREVIEWER COMMENTS

Reviewer #1 (Remarks to the Author):

The manuscript is well written but there are few notable points the authors must address:

1. The enthalpy guided alloy design strategy has been compared with the with the topological optimization approach in the introduction section. For nanodominated amorphous alloys there are many notable computational design approaches like multiscale modelling, Machine learning based modelling, evolutionary algorithms etc. Can the authors throw some light on the limitations of these approaches while comparing their own approach with advantages? Can a comparative analysis with work published in the literature be carried out in the introduction part?
2. What are the advantages of going by the bottom-up approach as compared to templated synthesis and high throughput experimentation?
3. Can the authors elaborate the sentence a bit more clearly- "Observed strain hardening of the hierarchically nanodominated amorphous alloy is not due to energy elevation", as mentioned in Page no.5 of the manuscript? Why would there be strain hardening in the nanodominated amorphous alloys due to energy elevation in the first place? Are there any structures showing strain hardening due to energy elevation in the literature?
4. The authors have mentioned positive mixing enthalpy for constituent elements to prevent inter diffusion between the neighboring amorphous nanodomains. But how does one consider calculating the magnitude of this positive enthalpy before selecting a combination of the elements? How to decide that a certain combination is chemically more heterogeneous than another combination as both may show positive enthalpy of mixing?
5. What is meant by a "buffering effect when tensile cracks eventually emerge", in the last paragraph of Page 10 of the manuscript? Is it related to the ductile mode of crack propagation as in case of the nano-bridging in contrast to the shear band cracking as shown by metallic glasses in general?
6. The authors may consider doing a more critical literature review of similar studies published in literature to provide a fine contrast with their current work highlighting notable achievements. The findings of the research work are of great relevance to designing amorphous structures overcoming the traditional problems encountered in localized shear bands formed in MGs and hence requires a critical analysis of previous work.

Reviewer #2 (Remarks to the Author):

The authors conducted in-situ compression and tensile tests to study the mechanical properties of a

hierarchically structured glass material composed of Ti-Zr-Nb-Si-XX/Mg-Zn-Ca-YY, and proposed new deformation mechanisms. While the idea of obtaining hierarchic structures is not new, the authors demonstrated superior mechanical behavior for this material. The paper requires major revisions based on the following comments:

- 1) Figure 2 only presents one compression test curve for each sample. More results should be presented to ensure statistical significance.
- 2) The terms Mono-MG and Hetero-MG need clarification.
- 3) The authors should explain the small serrations observed at the beginning of the stress-strain curves (around 5-10% strain) for the Ti-Zr-Nb-Si-XX/Mg-Zn-Ca-YY sample in Figure 2.
- 4) On page 5, in the paragraph "We checked the morphology of the deformed pillar with ~30% stain," the authors should correct the spelling of "strain."
- 5) Extended Data Figure 3 does not provide evidence of strain hardening. Instead, the figure shows that the deformed sample exhibits higher strength than the undeformed sample, likely due to a change in chemical composition and structure (The undeformed pillar consists of a layered structure of Mg- and Ti-based glasses. After deformation, apparently, there is no more a layered structure, as shown in Figure 3). The comparison between the samples is therefore not valid.
- 6) The authors should address whether the dynamic atomic intermixing observed during deformation could be attributed to the material's interaction with the electron beam, and provide evidence to support their claim.
- 7) The authors describe another combination of materials, Cu-Zr-Al + Ti-Zr-Nb-Si, in the methods section, but did not present any results. They should either remove this sentence or include the results of the Cu-Zr-Al + Ti-Zr-Nb-Si combination to evaluate its behavior.

Response to the Reviewers' Report

NCOMMS-23-04169-T

We would like to cordially thank the reviewers for the valuable suggestions and comments on our paper. Our response is structured as follows: The comments from the reviewers are copied below in black and italic font. For each comment, we present a detailed response, a full description of the newly added experimental results, and the corresponding manuscript modifications (blue font: reply items). The amended manuscript is enclosed, and the changes we made to the revised version are shown in red font (change items).

Referee #1:

“The manuscript is well written but there are few notable points the authors must address”.

Response: We are very grateful to the reviewer for the supportive comments. We have carefully addressed all the comments raised by the reviewer, as shown below in detail. We believe that the quality of the manuscript has been significantly improved in the revised version, thanks to the valuable suggestions provided by the reviewer.

“1. The enthalpy guided alloy design strategy has been compared with the topological optimization approach in the introduction section. For nanodomained amorphous alloys there are many notable computational design approaches like multiscale modelling, Machine learning based modelling, evolutionary algorithms etc. Can the authors throw some light on the limitations of these approaches while comparing their own approach with advantages? Can a comparative analysis with work published in the literature be carried out in the introduction part?”

Response: We are grateful to the reviewer for the constructive comments. We fully comply and have added the computational design references and the corresponding discussions in the revised manuscript.

Modifications: On page 2 of the revised manuscript, we have revised the following text:

Computational simulations have shown its advantages in predicting nanostructure of MGs with homogeneous plastic flow. Evolutionary algorithms⁹ are able to search for MG candidates with excellent glass forming ability (GFA), and thus optimize the existing MGs. Machine learning

based modellings¹⁰⁻¹² can predict structure heterogeneities for different MG systems and correlate their structures with mechanical properties. Multiscale modelling by molecular dynamics simulations^{13,14} demonstrated that heterogeneous MGs (nanoglass or nanolaminated composites) have enhanced homogeneous deformation. In experiments, however, it is difficult to achieve very large heterogeneity as that in simulations and the mechanical strain rates are much lower ($\sim 10^{-3} \text{ s}^{-1}$ vs $\sim 10^7 \text{ s}^{-1}$). Adding crystalline phase into amorphous matrix^{15,16} is an efficient experimental approach to enhance plasticity, resulting from dislocation movement inside the crystalline phase and multiple shear banding events in the amorphous matrix. By reducing the size of the amorphous phase to sub-10 nm in the crystal-glass alloys¹⁷⁻¹⁹, confined plastic flow of the nano-sized amorphous phase can be realized, due to intrinsic small size and much different mechanical responses of the crystalline and amorphous phases. Therefore, here we conceived the idea of introducing nanodomains with large mechanical difference in fully amorphous alloys, in order to realize confined plastic flow with no mature shear bands in the nanodomains.

Accordingly, we have added the following references:

- 9 Forrest, R. M. & Greer, A. L. Evolutionary design of machine-learning-predicted bulk metallic glasses. *Digital Discovery* **2**, 202 (2023).
- 10 Samavatian, M. *et al.* Characterization of nanoscale structural heterogeneity in metallic glasses: a machine learning study. *J. Non-Cryst Solids* **578**, 121344 (2022).
- 11 Wu, Y., Xu, B., Zhang, X. & Guan, P. Machine-learning inspired density-fluctuation model of local structural instability in metallic glasses. *Acta Mater.* **247**, 118741 (2023).
- 12 Fan, Z., Ding, J. & Ma, E. Machine learning bridges local static structure with multiple properties in metallic glasses. *Mater. Today* **40**, 48-62 (2020).
- 13 Sha, Z.-D., Branicio, P. S., Lee, H. P. & Tay, T. E. Strong and ductile nanolaminate composites combining metallic glasses and nanoglasses. *Int. J. Plast.* **90**, 231-241 (2017).
- 14 Adibi, S. *et al.* A transition from localized shear banding to homogeneous superplastic flow in nanoglass. *Appl. Phys. Lett.* **103**, 211905 (2013).
- 15 Hofmann, D. C. *et al.* Designing metallic glass matrix composites with high toughness and tensile ductility. *Nature* **451**, 1085-1089 (2008).
- 16 Wu, Y., Xiao, Y., Chen, G., Liu, C. T. & Lu, Z. Bulk metallic glass composites with transformation - mediated work - hardening and ductility. *Adv. Mater.* **22**, 2770-2773 (2010).
- 17 Khalajhedayati, A., Pan, Z. & Rupert, T. J. Manipulating the interfacial structure of nanomaterials to achieve a unique combination of strength and ductility. *Nat. Commun.* **7**, 10802 (2016).
- 18 Wu, G. *et al.* Hierarchical nanostructured aluminum alloy with ultrahigh strength and large plasticity. *Nat. Commun.* **10**, 5099 (2019).
- 19 Ming, K. *et al.* Enhancing strength and ductility via crystalline-amorphous nanoarchitectures in TiZr-based alloys. *Sci. Adv.* **8**, eabm2884 (2022).

“2. What are the advantages of going by the bottom-up approach as compared to templated synthesis and high throughput experimentation?”

Response: We are grateful to the reviewer for the comments. As suggested by the reviewer, we have compared the templated synthesis and high throughput experimentation with our approach in the revised manuscript.

Modifications: On page 3 of the revised manuscript, we have revised the following text:

Magnetron sputtering depositions with template substrates²¹ or co-sputtering depositions via high throughput method²² are efficient to fabricate materials with compositional gradient in horizontal direction, and thus different microstructures can be achieved. These strategies show their advantages in the development of compositional complex crystalline²³ and amorphous²⁴ alloys. The purpose of the current study is to fabricate materials with hierarchically nanodominated compositions in both horizontal and vertical directions, which is different from the above strategies.

Accordingly, we have added the following references:

- 21 Chang, H. *et al.* Combinatorial synthesis and high throughput evaluation of ferroelectric/dielectric thin-film libraries for microwave applications. *Appl. Phys. Lett.* **72**, 2185-2187 (1998).
- 22 Hanak, J. J. The “multiple-sample concept” in materials research: Synthesis, compositional analysis and testing of entire multicomponent systems. *J. Mater. Sci.* **5**, 964-971 (1970).
- 23 Yan, X. H., Li, J. S., Zhang, W. R. & Zhang, Y. A brief review of high-entropy films. *Mater. Chem. Phys.* **210**, 12-19 (2018).
- 24 Li, M.-X. *et al.* High-temperature bulk metallic glasses developed by combinatorial methods. *Nature* **569**, 99-103 (2019).

“3. Can the authors elaborate the sentence a bit more clearly- "Observed strain hardening of the hierarchically nanodominated amorphous alloy is not due to energy elevation", as mentioned in Page no.5 of the manuscript? Why would there be strain hardening in the nanodominated amorphous alloys due to energy elevation in the first place? Are there any structures showing strain hardening due to energy elevation in the literature?”

Response: We thank the reviewer for the comment. In the revised manuscript, we have explained the energy elevation mechanism of the hierarchically nanodominated amorphous alloy and discussed energy elevation induced strain hardening based on the literature.

Modifications: On page 6 of the revised manuscript, we have revised the following text:

It is reported that MGs can be rejuvenated to an elevated energy state by confined deformation³⁶, which facilitates homogeneous plastic flow of ~1% strain (with strain hardening) in compression. It is known that the elevated energy³⁶ of MGs corresponds to ultra-fast cooling at ~10⁶ K/s, three to four orders of magnitude higher than the possible cooling rate for a conventional cast rod, but that effective cooling rate is lower than what we used in magnetron sputtering deposition (~10¹⁰ K/s)³⁷. Therefore, it is supposed that the deposited hierarchically nanodomained amorphous alloy has an elevated energy. However, the as-deposited Mg-Zn-Ca and Ti-Zr-Nb-Si reference amorphous alloys show shear banding dominated plastic deformation, which therefore indicates that the large homogeneous plastic flow and the observed strain hardening of the hierarchically nanodomained amorphous alloy is not due to energy elevation.

References

- 36 Pan, J., Ivanov, Y. P., Zhou, W., Li, Y. & Greer, A. Strain-hardening and suppression of shear-banding in rejuvenated bulk metallic glass. *Nature* **578**, 559-562 (2020).

“4. The authors have mentioned positive mixing enthalpy for constituent elements to prevent inter diffusion between the neighboring amorphous nanodomains. But how does one consider calculating the magnitude of this positive enthalpy before selecting a combination of the elements? How to decide that a certain combination is chemically more heterogeneous than another combination as both may show positive enthalpy of mixing?”

Response: We thank the reviewer for the comment. We have thoroughly revised the manuscript to clearly discuss the positive mixing enthalpy guideline for achieving nanodomains in amorphous alloys, in full accordance to the reviewer’s comments.

Modifications: On Page 3 of the revised manuscript, we have revised the following text:

The constituent elements in each nanodomain has large negative mixing enthalpy to guarantee amorphous structure, while the principal constituent elements from the two nanodomains have positive mixing enthalpy, reducing the driving force for interdiffusion²⁵. Although positive mixing enthalpy is required for the principal constituent elements of the neighboring amorphous nanodomains, the value of the mixing enthalpy should not be large, which sets an alloy design guideline. The hierarchically nanodomained amorphous alloy reveals atomic intermixing during plastic deformation, as shown in the context below. In this circumstance, the deformed alloy will suffer from potential crystallization if the positive mixing enthalpy of the principal constituent

elements were large. Therefore, small positive mixing enthalpy is needed. Here, the value of small positive mixing enthalpy should be in the range of 5-20 kJ/mol, as suggested in Ref.⁵. We summarized mixing enthalpy of atomic pairs²⁶ from frequently-used metallic elements (Extended Data Table 1), and highlighted the small positive values in green color, suggesting potential principal constituent elements for the neighboring amorphous nanodomains. For example, Ti and Mg have a small positive mixing enthalpy of 16 kJ/mol, and thus Ti-based and Mg-based amorphous alloys are good candidates for the neighboring amorphous nanodomains.

References

- 5 Kim, D., Kim, W., Park, E., Mattern, N. & Eckert, J. Phase separation in metallic glasses. *Prog. Mater. Sci.* **58**, 1103-1172 (2013).
- 26 Takeuchi, A. & Inoue, A. Classification of bulk metallic glasses by atomic size difference, heat of mixing and period of constituent elements and its application to characterization of the main alloying element. *Mater. Trans.* **46**, 2817-2829 (2005).

Accordingly, we added an Extended Data Table 1 on Page 19:

Extended Data Table 1 | The values of mixing enthalpy²⁶ (kJ/mol) calculated by Miedema's model for atomic pairs of frequently-used metallic elements. The mixing enthalpy with small positive values (5-20 kJ/mol) are highlighted in green color.

	Mg	Al	Ca	Ti	Fe	Co	Ni	Cu	Zn	Zr	Pd	Ag	Ta	W	Ir	Pt	Au
Au	-32	-22	-60	-47	8	7	7	-9	-16	-74	0	-6	-32	12	13	4	
Pt	-35	-44	-55	-74	-13	-7	-5	-12	-29	-100	2	-1	-66	-20	0		
Ir	-13	-30	-23	-57	-9	-3	-2	0	-13	-76	6	16	-52	-16			
W	38	-2	57	-6	0	-1	-3	22	15	-9	-6	43	-7				
Ta	30	-19	60	1	-15	-24	-29	2	-3	3	-52	15					
Ag	-10	-4	-28	-2	28	19	15	2	-4	-20	-7						
Pd	-40	-46	-63	-65	-4	-1	0	-14	-33	-91							
Zr	6	-44	37	0	-25	-41	-49	-23	-29								
Zn	-4	1	-22	-15	4	-5	-9	1									
Cu	-3	-1	-13	-9	13	6	4										
Ni	-4	-22	-7	-35	-2	0											
Co	3	-19	2	-28	-1												
Fe	18	-11	25	-17													
Ti	16	-30	43														
Ca	-6	-20															

Al	-2																
Mg																	

“5. What is meant by a “buffering effect when tensile cracks eventually emerge”, in the last paragraph of Page 10 of the manuscript? Is it related to the ductile mode of crack propagation as in case of the nano-bridging in contrast to the shear band cracking as shown by metallic glasses in general?”

Response: We thank the reviewer for the constructive comment. We fully comply with the reviewer’s suggestion “it is related to the ductile mode of crack propagation as in case of the nano-bridging”, and have revised the manuscript accordingly.

Modifications: On Page 11 of the revised manuscript, we have revised the following text:

Also, the design of hierarchical nanostructure successfully realizes a buffering effect by nano-bridging of the tensile cracks, thanks to the extensive homogeneous flow.

“6. The authors may consider doing a more critical literature review of similar studies published in literature to provide a fine contrast with their current work highlighting notable achievements. The findings of the research work are of great relevance to designing amorphous structures overcoming the traditional problems encountered in localized shear bands formed in MGs and hence requires a critical analysis of previous work.”

Response: We thank the reviewer for the suggestion. We fully comply and added the literature review accordingly.

Modifications: On Page 2 of the revised manuscript, we have revised the following text:

Computational simulations have shown its advantages in predicting nanostructure of MGs with homogeneous plastic flow. Evolutionary algorithms⁹ are able to search for MG candidates with excellent glass forming ability (GFA), and thus optimize the existing MGs. Machine learning based modellings¹⁰⁻¹² can predict structure heterogeneities for different MG systems and correlate their structures with mechanical properties. Multiscale modelling by molecular dynamics simulations^{13,14} demonstrated that heterogeneous MGs (nanoglass or nanolaminated composites) have enhanced homogeneous deformation. In experiments, however, it is difficult to achieve very large heterogeneity as that in simulations and the mechanical strain rates are much lower ($\sim 10^{-3} \text{ s}^{-1}$ vs $\sim 10^7 \text{ s}^{-1}$). Adding crystalline phase into amorphous matrix^{15,16} is an efficient

experimental approach to enhance plasticity, resulting from dislocation movement inside the crystalline phase and multiple shear banding events in the amorphous matrix. By reducing the size of the amorphous phase to sub-10 nm in the crystal-glass alloys¹⁷⁻¹⁹, confined plastic flow of the nano-sized amorphous phase can be realized, due to intrinsic small size and much different mechanical responses of the crystalline and amorphous phases. Therefore, here we conceived the idea of introducing nanodomains with large mechanical difference in fully amorphous alloys, in order to realize confined plastic flow with no mature shear bands in the nanodomains.

Accordingly, we have added the following references:

- 9 Forrest, R. M. & Greer, A. L. Evolutionary design of machine-learning-predicted bulk metallic glasses. *Digital Discovery* **2**, 202 (2023).
- 10 Samavatian, M. *et al.* Characterization of nanoscale structural heterogeneity in metallic glasses: a machine learning study. *J. Non-Cryst Solids* **578**, 121344 (2022).
- 11 Wu, Y., Xu, B., Zhang, X. & Guan, P. Machine-learning inspired density-fluctuation model of local structural instability in metallic glasses. *Acta Mater.* **247**, 118741 (2023).
- 12 Fan, Z., Ding, J. & Ma, E. Machine learning bridges local static structure with multiple properties in metallic glasses. *Mater. Today* **40**, 48-62 (2020).
- 13 Sha, Z.-D., Branicio, P. S., Lee, H. P. & Tay, T. E. Strong and ductile nanolaminate composites combining metallic glasses and nanoglasses. *Int. J. Plast.* **90**, 231-241 (2017).
- 14 Adibi, S. *et al.* A transition from localized shear banding to homogeneous superplastic flow in nanoglass. *Appl. Phys. Lett.* **103**, 211905 (2013).
- 15 Hofmann, D. C. *et al.* Designing metallic glass matrix composites with high toughness and tensile ductility. *Nature* **451**, 1085-1089 (2008).
- 16 Wu, Y., Xiao, Y., Chen, G., Liu, C. T. & Lu, Z. Bulk metallic glass composites with transformation - mediated work - hardening and ductility. *Adv. Mater.* **22**, 2770-2773 (2010).
- 17 Khalajhedayati, A., Pan, Z. & Rupert, T. J. Manipulating the interfacial structure of nanomaterials to achieve a unique combination of strength and ductility. *Nat. Commun.* **7**, 10802 (2016).
- 18 Wu, G. *et al.* Hierarchical nanostructured aluminum alloy with ultrahigh strength and large plasticity. *Nat. Commun.* **10**, 5099 (2019).
- 19 Ming, K. *et al.* Enhancing strength and ductility via crystalline-amorphous nanoarchitectures in TiZr-based alloys. *Sci. Adv.* **8**, eabm2884 (2022).

Referee #2:

“The authors conducted in-situ compression and tensile tests to study the mechanical properties of a hierarchically structured glass material composed of Ti-Zr-Nb-Si-XX/Mg-Zn-Ca-YY, and proposed new deformation mechanisms. While the idea of obtaining hierarchic structures is not new, the authors demonstrated superior mechanical behavior for this material.”

Response: We are grateful to the reviewer for recognition of the novelty of our hierarchically structured glass material. We also thank the reviewer for recognition of the superior mechanical behavior and the new deformation mechanisms. We have carefully addressed all the comments raised by the reviewer, as shown below in detail. We believe that the quality of the manuscript has been significantly improved in the revised version, thanks to the valuable suggestions provided by the reviewer.

“The paper requires major revisions based on the following comments:

1) Figure 2 only presents one compression test curve for each sample. More results should be presented to ensure statistical significance.”

Response: We thank the reviewer for the valuable comments. We fully comply and added more results to ensure statistical significance.

Modifications: On Page 6 of the revised manuscript, we have revised the following text:

The pillar compression tests under identical conditions were repeated for at least 3 times to ensure statistical significance (Extended Data Figure 3).

Accordingly, on Page 22 of the revised manuscript, we added an Extended Data Figure 3:

Extended Data Figure 3 | Mechanical properties of the hierarchically nanodominated amorphous alloy, Mg-Zn-Ca amorphous alloy and Ti-Zr-Nb-Si amorphous alloy. a-c, Stress-strain curves of the alloys. The pillar compression tests under identical conditions were repeated for at least 3 times. **d,** Yield strength of the alloys.

“2) The terms Mono-MG and Hetero-MG need clarification.”

Response: We thank the reviewer for the suggestion. We have added clarification of the Mono-MG and Hetero-MG in the revised manuscript.

Modifications: On Page 7 of the revised manuscript, we have revised the following text:

Figure 2 | Superior mechanical property of the hierarchically nanodomained amorphous alloy compared with that of reference mono- and hetero-MGs. Here, mono- and hetero-MGs represent MGs with homogeneous and heterogeneous structures, respectively.

“3) The authors should explain the small serrations observed at the beginning of the stress-strain curves (around 5-10% strain) for the Ti-Zr-Nb-Si-XX/Mg-Zn-Ca-YY sample in Figure 2.”

Response: We thank the reviewer for the comment. We fully comply and explained the origin of small serrations observed at the beginning of the stress-strain curves (around 5-10% strain) for the Ti-Zr-Nb-Si-XX/Mg-Zn-Ca-YY sample in Figure 2.

Modifications: On Page 9-10 of the revised manuscript, we have revised the following text:

The Young's modulus of the Mg-Zn-Ca-YY domain is expected to be smaller than that of the Ti-Zr-Nb-Si-XX domain, considering that the Young's modulus of the Mg-Zn-Ca and Ti-Zr-Nb-Si reference amorphous alloys is 45 GPa and 110 GPa, respectively, obtained from nanoindentation. Therefore, **the profuse ST events occur earlier in Mg-Zn-Ca-YY domain, and as the flow stress increasing, these events occur in the Ti-Zr-Nb-Si-XX domain afterwards.** The non-homogeneous deformation⁴⁵ of the two amorphous domains **induces the small serrated stress flow during plastic deformation.**

References

45 Zhu, Y. & Wu, X. Perspective on hetero-deformation induced (HDI) hardening and back stress. *Mater. Res. Lett.* **7**, 393-398 (2019).

Accordingly, on Page 6 of the revised manuscript, we have revised the following text:

We checked the morphology of the deformed pillar with ~30% **strain**, and it shows no shear bands (inset, Figure 2a). Therefore, the **stress deviation (apart from the green dashed line on Figure 2a)** and small serrations on the stress-strain curve after 40% strain correspond to multiple shear banding events.

“4) On page 5, in the paragraph "We checked the morphology of the deformed pillar with ~30% stain," the authors should correct the spelling of "strain."”

Response: We thank the reviewer for the comment. We have revised the typo accordingly.

Modifications: On Page 6 of the revised manuscript, we have revised the following text:

We checked the morphology of the deformed pillar with ~30% **strain**

“5) Extended Data Figure 3 does not provide evidence of strain hardening. Instead, the figure shows that the deformed sample exhibits higher strength than the undeformed sample, likely due to a change in chemical composition and structure (The undeformed pillar consists of a layered structure of Mg- and Ti-based glasses. After deformation, apparently, there is no more a layered structure, as shown in Figure 3). The comparison between the samples is therefore not valid.”

Response: We thank the reviewer for the valuable suggestion. We fully comply and have revised the manuscript accordingly.

Modifications: On Page 6 of the revised manuscript, we have revised the following text:

The yield strength of the pre-deformed material is 1.8 GPa, 0.2 GPa higher than that of the undeformed material, **due to a change in chemical composition and structure (atomic intermixing between the two nanodomains).**

On Page 23 of the revised manuscript, we have also revised the figure caption accordingly:

Extended Data Figure 4 | Yield strength increase of the hierarchically nanodominated amorphous alloy after deformation.

“6) The authors should address whether the dynamic atomic intermixing observed during deformation could be attributed to the material's interaction with the electron beam, and provide evidence to support their claim.”

Response: We thank the reviewer for the comment. The composition analyses were conducted on samples after *ex-situ* deformation in air, instead of SEM/TEM *in-situ* deformation. Therefore, the electron beam induced composition and structure changes can be ruled out during compression. We are sorry we did not make it clear in the original manuscript. We have revised the manuscript to explain the experimental conditions more clearly.

Modifications: On Page 17 of the revised manuscript, we have revised the following text:

The Hysitron TI950 nanoindenter is an *ex-situ* instrument in air, and thus the electron beam induced composition and structure changes can be ruled out during compression.

“7) The authors describe another combination of materials, Cu-Zr-Al + Ti-Zr-Nb-Si, in the methods section, but did not present any results. They should either remove this sentence or include the results of the Cu-Zr-Al + Ti-Zr-Nb-Si combination to evaluate its behavior.”

Response: We thank the reviewer for the suggestion. We fully comply and have removed this sentence and the corresponding Extended Figure.

REVIEWERS' COMMENTS

Reviewer #1 (Remarks to the Author):

The manuscript can be accepted in present form

Reviewer #2 (Remarks to the Author):

The authors have made all the necessary corrections and modifications as indicated, so I recommend approving the manuscript.

Response to the Reviewers' Report

NCOMMS-23-04169-A

REVIEWERS' COMMENTS

Reviewer #1 (Remarks to the Author): The manuscript can be accepted in present form

Response: We are very grateful to the reviewer for the supportive comments.

Reviewer #2 (Remarks to the Author): The authors have made all the necessary corrections and modifications as indicated, so I recommend approving the manuscript.

Response: We are very grateful to the reviewer for the supportive comments.